# New Approaches to Modelling Wilderness Quality in Iceland

**Steve Carver** [1,*] **, Sif Konráðsdóttir** [2] **, Snæbjörn Guðmundsson** [3] **, Ben Carver** [4] **and Oliver Kenyon** [4]

1    School of Geography, University of Leeds, Leeds LS2 9JT, UK
2    Attorney-at-Law, ÓFEIG Nature Conservation Society, 107 Reykjavík, Iceland
3    Icelandic Museum of Natural History, 108 Reykjavík, Iceland
4    Wildland Research Limited, Northallerton DL7 8FF, UK
*    Correspondence: s.j.carver@leeds.ac.uk; Tel.: +44-(0)113-3433318

**Abstract:** Much of Europe's remaining wilderness areas are found in Iceland, yet few are formally protected despite ongoing threats from renewable energy exploitation and $4 \times 4$ usage. Robust and repeatable approaches are required to map wilderness landscape qualities in support of developing policy on designations that meet international standards. We present an approach to mapping wilderness that is based on internationally recognised methods and customised to suit the unique nature of Icelandic landscapes. We use spatially explicit models of wilderness attributes that measure human impact from vehicular access, land use and visible human features rather than relying on proxy measures such as buffer zones. Seventeen wilderness areas are identified across the Central Highlands and surrounding areas, totalling some 28,470 km$^2$. These are compared to existing mapping projects. The character of these areas is described using additional spatial data models on openness, ruggedness and accessibility from settlements, together with information on mobile phone coverage and grazing patterns. This is the most detailed mapping of wilderness in Iceland to date and an important step towards the formal definition of boundaries of wilderness areas meeting IUCN Category 1b and Wild Europe Working Definition in Iceland.

**Keywords:** wilderness quality; wilderness character; Iceland; Central Highlands

## 1. Introduction

Wilderness is an increasingly rare landscape resource characterised by the IUCN as "protected areas that are usually large, unmodified or slightly modified areas, retaining their natural character and influence, without permanent or significant human habitation, which are protected and managed so as to preserve their natural condition" [1] (p. ii). Recent research using global datasets has highlighted alarming rates of loss with estimates ranging from a nearly 10% loss between 1993 and 2009 [2] to 175 km$^2$ of wilderness lost per day [3], most of it due to land-take for agriculture and urban expansion [4]. These rapid rates of attrition comprise a principal threat to biodiversity conservation and UN Sustainable Development Goals [5] such that the post-2020 Global Biodiversity Framework of the Convention on Biological Diversity has placed "retaining wilderness areas" as the first of 21 action-oriented targets for 2030 [6].

The European Parliament recognised the importance of protecting Europe's wilderness areas in February 2009 with a subsequent policy paper calling for wilderness to be defined, mapped, and protected at all levels [7]. The resulting guideline document on wilderness within the Natura 2000 protected area network refines the definition of wilderness in Europe as "an area governed by natural processes . . . composed of native habitats and species, and large enough for the effective ecological functioning of natural processes. It is unmodified or only slightly modified and without intrusive or extractive human activity, settlements, infrastructure, or visual disturbance" [8], p. 10. An EU wilderness register and map published in 2013 highlighted disparities in wilderness protection across Europe. This revealed interesting patterns in remaining wilderness within EU states and

partner countries based on the mapping of potential naturalness of vegetation, remoteness from settlements and other human infrastructure and remoteness from roads [9]. This work shows clear altitudinal and latitudinal trends in these data with most of Europe's wildest landscapes being found in high-latitude (Arctic and near-Arctic) and high-altitude (mountainous) areas. Other interesting trends are seen in the level of protection afforded to the mapped wilderness with many large areas, particularly in northern Scandinavia and Iceland, remaining unprotected despite possessing all the attributes of wilderness [10].

Retaining wilderness is one of the stated objectives of the Icelandic Nature Conservation Act No 60/2013 (NCA), Article 3. Article 5 (19) of the Act defines wilderness as "An uninhabited area that is in principle at least 25 km$^2$ in size or in such a way that one can enjoy solitude and nature there without disturbance from man-made structures or the traffic of motorized vehicles and in principle at least 5 km away from structures and other technical traces, such as power lines, power plants, reservoirs and built roads" [11]. While different from those definitions provided by the EU and IUCN, this is closely linked to the conditions for designating lands as wilderness protected areas given in Article 46, which states to retain the wilderness as "Large areas, in principle untouched by human activities, where nature can evolve independently, may be legally designated as wilderness protected areas" and that "The designation shall aim at protecting the characteristics of the areas, for example to maintain diverse and unique landscape, openness and/or protecting large ecosystems; and to ensure that present and future generations can enjoy solitude and the nature without disturbance from man-made structures or the traffic of motorized vehicles" [11].

These provisions were a novelty when the nature conservation law reform entered into force late 2015, with the preparatory legislative work referring explicitly to IUCN Category 1b for wilderness designation. However, to date, no area within the Central Highlands has been designated as a wilderness protected area despite provision for doing so within the NCA. A more recent legal novelty, entering into force early 2021 and adding Article 73a, together with a temporary provision to the NCA, provides for the mapping of the wilderness areas across Iceland "in line with internationally recognized methodology" [11]. The work presented here was initiated locally and developed by the paper's authors against this legal background.

Iceland is a unique and important case as regards wilderness in Europe and as such is worth careful attention. The work of Kuiters et al. [9] shows that as much as 43% of Europe's top 1% wildest areas fall within Iceland, and as such, Iceland represents a significant resource for nature protection as well as tourism and recreation [12]. While much of this presents as the extensive icecaps of the Vatnajökull, Hofsjökull, etc., large areas of the Central Highlands comprising ice-free hills, mountains, rivers, lakes and expansive gravel plains are also included in the 43% figure. The fact that many of these areas are currently unprotected highlights the need for appropriate and locally specific methods to assist the authorities in identifying variations in wildness across Iceland building on the IUCN, European and Icelandic definitions of wilderness as stated in the text of the 2013 NCA and subsequent amendments. An Iceland-specific approach to modelling wilderness quality that builds on existing recognised methods is therefore required to identify boundaries of wilderness areas for designation and ensure future protection. Such methods are needed to support the planning process through strategic and responsive "what if?" modelling of proposed developments (e.g., renewable energy projects) to reliably predict and illustrate the likely impacts should they go ahead [13].

Different countries and their local cultures often project different understandings of what is meant by "wilderness" and what it means for landscapes and protected areas. In Iceland, óbyggð víðerni (usually shortened to víðerni) is used as a legal term, which literally translated means "uninhabited wilderness". This corresponds broadly to IUCN Category 1b areas. However, in local vernacular, it is usual to use words such as óbyggðir (literally meaning "uninhabited area") and miðhálendi (as a place term referring specifically to the uninhabited areas of the Central Highlands) [14]. Words aside, much of Iceland's interior

landscapes may reliably and reasonably be classified as wilderness once away from roads and influences from other human infrastructure and land use.

The landscape of Iceland's interior is unique within Europe, and perhaps the rest of the world. It is characterised by a spectacular mix of glaciers and icecaps, wide flat gravel plains (or 'sandurs'), rolling hills and rugged mountains interspersed with glacier-fed rivers, hot springs, and deep valleys [15]. The overall impression is of a primeval, almost moon-like landscape shaped entirely by the forces of nature. Geologically speaking, Iceland is young (the oldest exposed rocks are approximately 15–16 million years old) with volcanic landforms of lava flows, cinder cones, geothermal areas and active volcanoes as key characteristics along the volcanic rift zone of the Central Highlands [16]. Water, either in the form of snow and ice or huge glacial rivers, lakes, ponds and springs, is also a key element that provides interest and often forms a barrier to movement, thus increasing remoteness. Vegetation is often sparse or non-existent with Arctic/Alpine plant communities and moss carpets dominating, with its low stature creating an open landscape feel across much of the interior. Example landscapes of the Central Highlands are shown in Figure 1.

In this paper, we develop an Iceland-specific approach to modelling wilderness quality as a basis for robust mapping wilderness boundaries; the principal aim being to support the Icelandic government in their designation process in meeting both the objectives of the NCA (2013) and UN Sustainable Development goals. The specific objectives of the paper are to: (a) modify existing and recognised wilderness quality models to create a custom approach suitable for the Icelandic landscape; (b) apply IUCN and European wilderness definitions and criteria to define existing wilderness areas and map their boundaries; and (c) describe the wilderness character of the resulting areas based on additional spatial attributes. We propose a 4Rs approach utilising:

1.  Rigorous, spatially explicit models of attributes influencing wilderness quality;
2.  Robust measurement of wilderness attributes describing human landscape impacts such as remoteness (time taken to walk from nearest point of mechanised access), visual impact (proportion of the landscape occupied by human features), and land use (affecting perceived naturalness of ecosystems);
3.  Repeatable analyses that can achieve the same results each time the model is run enabling accurate predictions of impacts from proposed developments and associated changes in wilderness quality; and
4.  Reliable interpretations of wilderness definitions using best available data at high enough resolutions enabling comparability of work at both local and national scales.

Previous mapping work has tended to focus primarily on the size and distance thresholds outlined in the NCA and previous versions of the wilderness definition. While some attempts have been made at visual impact analysis, the resulting maps interpret the more objective part of the definition of wilderness from the NCA using simple buffers to identify areas at least 3 or 5 km away from roads, buildings, and other human infrastructure, and then reselecting those resulting areas that are at least 25 km$^2$ in size [17]. One exception has been the innovative use of Participatory GIS (PGIS) by Ólafsdóttir and Sæþórsdóttir [18] to compare these areas with crowd-sourced perceptions of wilderness among local people and tourists. Here, an online map is used together with a spray can tool (Map-Me) to allow users to define their own wilderness areas by spraying directly on the map [19].

We suggest that buffer zones and reselections based on the distance and area thresholds alone, as taken from the objective part of the NCA definition, are proxy measures and do not measure actual impacts associated with human infrastructure within the Central Highlands. As such, these fail to capture the core of the wilderness definition as intended by the legislator. The application of such proxy measures needs to be carried out with care, as the results can be misleading. For example, a rough, single-track gravel road can have the same effect as a paved and elevated dual carriageway road, whereas its true impact is dependent on its type (and traffic volume), how visible it is and how long it takes to walk from it into the surrounding landscape. Weighted buffer zones using different buffer widths to account for road type and traffic volume can go some way towards estimating variations

in the degree of impact [20] but cannot accurately measure impacts in terms of naturalness, visibility and remoteness. Other uncertainties and differences can be further introduced in deciding which roads to include in the mapping exercise. Ostman et al. [20] exclude all unpaved gravel roads from their maps with the result that the size and extent of wilderness areas within the Central Highlands are greatly over-estimated despite these roads having a similar impact to paved roads, at least in terms of remoteness from motorized access. This is inconsistent with the legal text and interpretation of the NCA definition itself, and furthermore, such a categorisation of roads is not supported by the NCA's reference to IUCN Category 1b criteria.

Rather than rely on proxy measures, we develop an Iceland-specific approach to modelling impacts from human infrastructure and land use on wilderness quality that is based on the actual measurement of these impacts using spatial interpretations of the EU and IUCN wilderness definitions as suggested in the preparatory work of the NCA. Our approach is based on the legal interpretation of reformed Icelandic law in the field of nature conservation and wilderness as described above. Our research builds on existing, internationally recognised methods, as suggested in the latest amendments to the NCA, and applies these to Iceland with regard to the characteristics of the Central Highlands landscape. Existing examples include mapping wildness in Scottish National Parks [21,22] and wild land areas (WLAs) across Scotland by Scottish Natural Heritage [23]; mapping Haute Naturalité, or high naturalness, across France for IUCN France [24]; mapping variations in wilderness characteristics in designated wilderness areas for the US National Park Service [25]; and modelling variations in wilderness quality across China [26]. Adapting and enhancing these approaches enables us to model impacts on wilderness quality with reference to the 4Rs and then apply EU and IUCN wilderness definitions to draw wilderness protected area boundaries and describe their character. The resulting models represent a more rigorous, robust, and reliable representation of actual patterns of wilderness than those achievable using proxy measures and a tool with which the impacts of proposed future developments and planning decisions can be accurately predicted through repeat mapping.

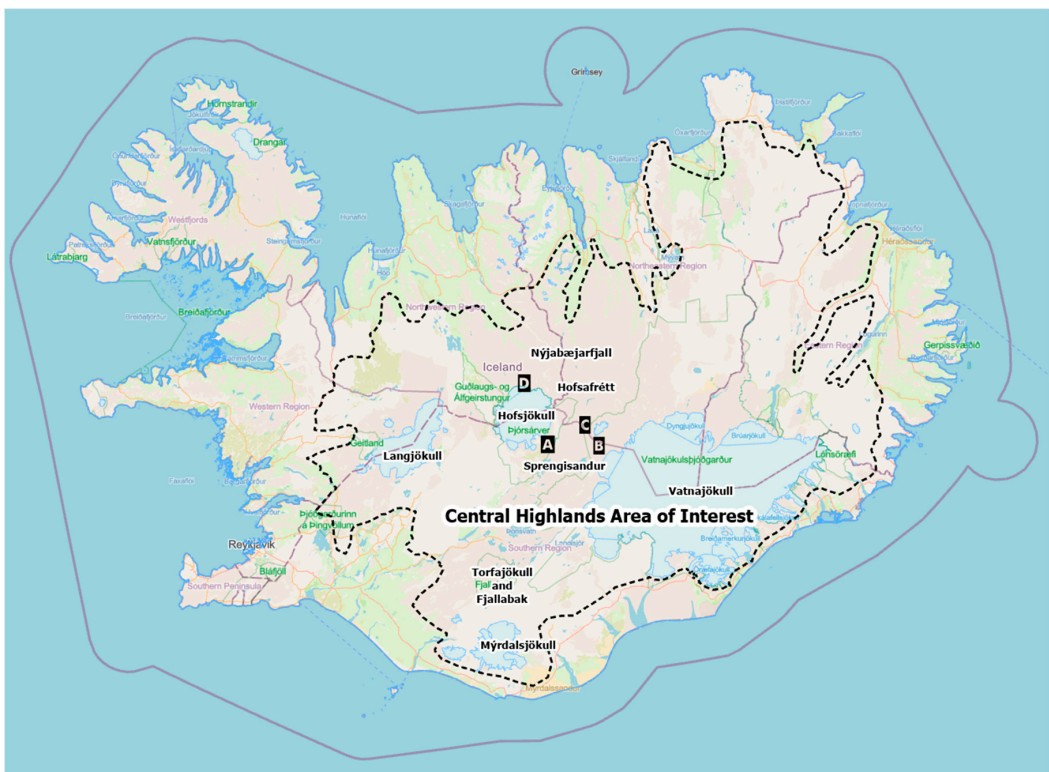

**Figure 1.** *Cont.*

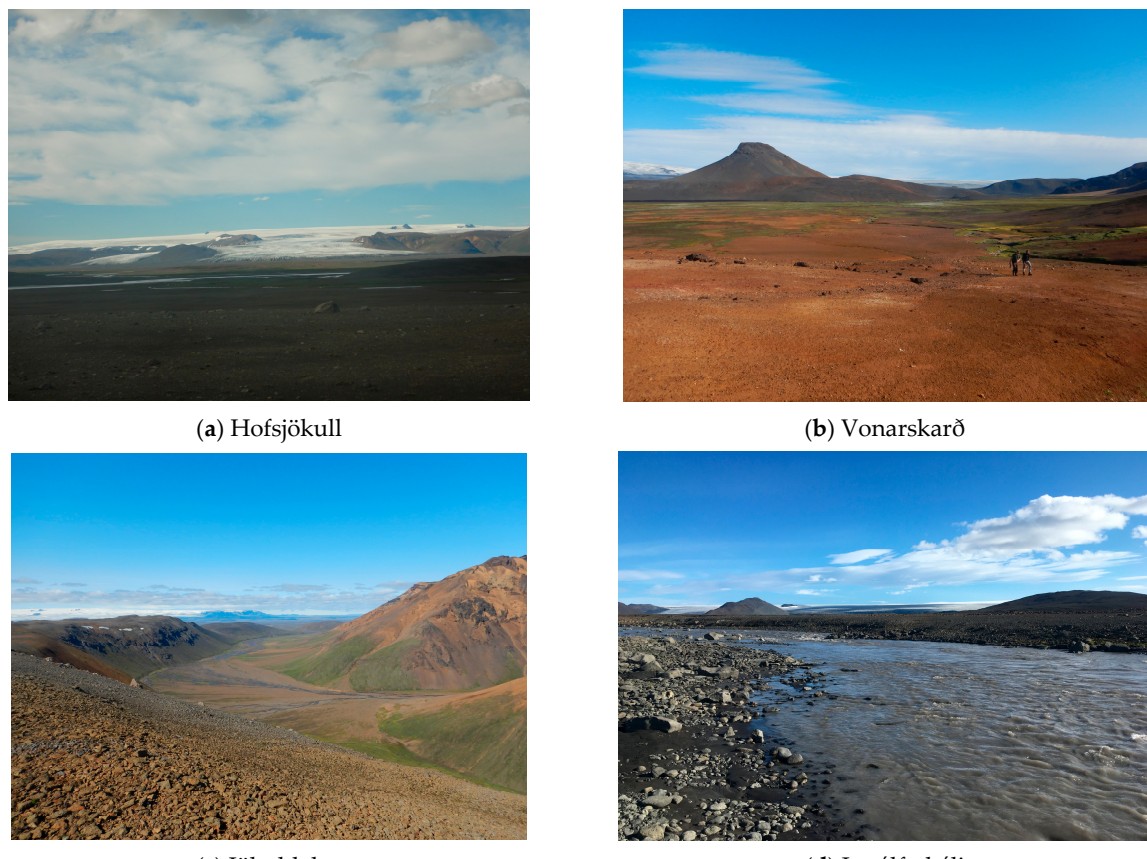

(**a**) Hofsjökull

(**b**) Vonarskarð

(**c**) Jökuldalur

(**d**) Ingólfsskáli

**Figure 1.** Map and landscapes of the Central Highlands: (**a**) Hofsjökull, (**b**) Vonarskarð, (**c**) Jökuldalur, (**d**) Ingólfsskáli.

## 2. Materials and Methods

A simple approach to modelling wilderness quality in Iceland would be to just apply one of the existing methods such as that employed in Scotland [22]. However, the variety seen in surface form and geographical context within the Central Highlands of Iceland creates the need for a two-part model that can firstly model variations in wilderness quality and secondly categorise individual areas depending on their landscape character and those human features affecting public perceptions of wilderness.

The first part of the method is a more traditional "Wilderness Quality Index" (WQI), based on a multi-criteria evaluation (MCE) of three principal attributes: (1) remoteness from mechanised access (or time taken to walk from a motorised vehicle); (2) lack of visual intrusion from modern human artefacts; and (3) perceived naturalness of land cover. When used together, these key attributes can model the spatial variation in wilderness quality, which can then help define wilderness core, buffer and transition zones by careful application of appropriate size and areas thresholds derived from EU and IUCN wilderness definitions. The second part of this model focuses on wilderness character using additional spatial datasets to describe, map and tabulate the unique characteristics of the areas defined in part 1 of the method. This includes further detail from spatial models of openness, ruggedness and accessibility (time taken to drive from human settlements) and additional information provided from maps of mobile phone coverage, livestock grazing and broader landscape character assessments. This two-part method provides detail and nuance in the mapping of key attributes and overall wilderness quality while providing further information about the character of each of the resulting core wilderness areas, thus meeting the need for a reliable, rigorous, robust and repeatable method that can be confidently used to inform decisions about policy on protected areas. This is summarised in Figure 2.

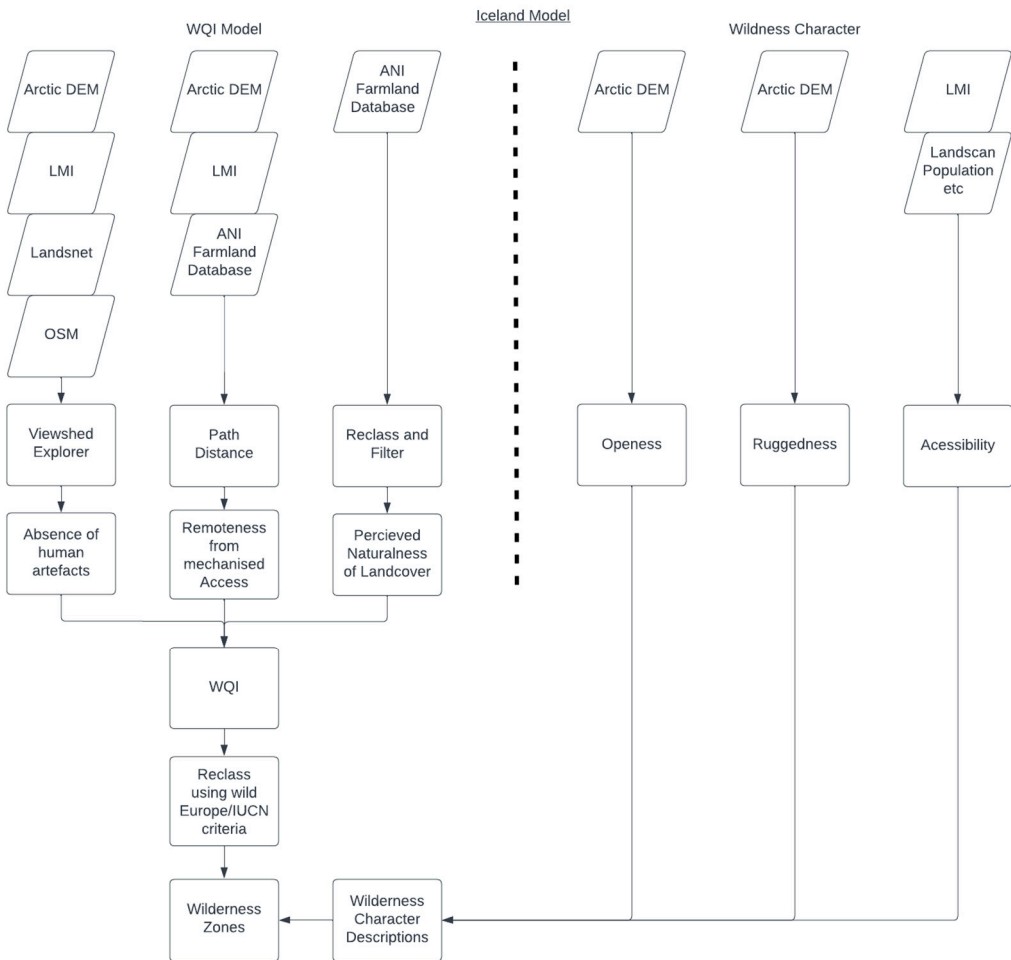

**Figure 2.** Model flow chart.

## 2.1. Method Development

Earlier work on wilderness quality mapping by Lesslie and Maslen [27] for the Australian National Wilderness Inventory (ANWI) and adapted by Carver et al. [22] for Scotland's national parks uses four wilderness attributes to create a combined map of wilderness or a WQI. Many wild areas are often characterised by their rugged nature (thus limiting their utility), but this is not always the case, leading to bias in mapped wildness towards mountainous areas or rugged coastlines. For example, in the Scottish wild land mapping, areas such as the low-lying Flow Country in the far northeast of the Scottish mainland are under-represented due to the flat nature of the terrain, despite this landscape being extremely challenging and difficult to cross due to its boggy nature. This is true also for Iceland's Central Highlands, where wilderness areas span a range of landscape types from the many wide open gravel plains such as Sprengisandur and Hofsafrétt, and icecaps including the Vatnajökull and Hofsjökull, while enclosed and rugged valleys are found locally in other areas such as Nýjabæjarfjall in the north and Torfajökull/Fjallabak area in the south (see Figure 1). Variations in topography thus have a marked influence on sense of space and openness as well as impacting on patterns of visual impact and remoteness.

To control for this, the attributes used to map wilderness quality are restricted to remoteness from mechanised access, absence of modern human artefacts, and perceived naturalness of land cover, thereby avoiding possible bias by inclusion of a ruggedness layer in this part of the method. These attributes, together with the data sources and approaches used to map them, are described in Section 2.2 below.

Potential wilderness areas are defined by classifying the resulting WQI into interior core, core, buffer, transition, and non-wild zones using statistical methods. Here, a Jenks

Natural Breaks model is applied as per the Scottish WLA mapping [23]. The size and area thresholds from the Wild Europe Working Definition for wilderness [28] are then applied to these zones to produce a set of wilderness area boundaries meeting the criteria from IUCN Category 1b guidelines and the NCA definition.

These areas are then described using additional information (including openness, ruggedness, accessibility to centres of population, etc.) to create individual maps and tabulate wilderness character descriptions building on the work and experience of the US National Park Service 'Keeping It Wild' wilderness character mapping [25].

*2.2. Wilderness Attributes*

Three attributes are used to model spatial patterns of wilderness and create a WQI for the Central Highlands area. Justification for their inclusion, data sources, models used, and outputs are described for each attribute below.

2.2.1. Remoteness from Mechanised Access

Remoteness is a key element determining wilderness quality since it affects how a human subject feels being separated from the modern world and our mechanical modes of transportation and also reflects both the effort required to obtain a location by non-mechanical means and personal risk/safety should something go wrong (e.g., injury or bad weather). Remoteness is modelled here using Naismith's Rule [29] as described by Carver et al. [22] for mapping wildness in Scotland's National Parks. Given the varied and challenging nature of the terrain found within the Central Highlands, it is essential to include terrain as a principal variable governing remoteness across the area. A GIS implementation of Naismith's Rule used here incorporates detailed terrain and land cover information to estimate the time in seconds required to walk from the nearest point of mechanised access, be that a paved road or gravel track, taking the effects of distance, relative slope, ground cover and barrier features such as open water, large rivers, crevassed areas of icecaps and very steep ground into account. This assumes remoteness to be directly proportional to the time taken to walk from the nearest road across varied terrain and land cover types. This is performed in ArcGIS Pro 3.0 using the Distance Accumulation tools. The implementation of this model of remoteness requires a detailed terrain model and ancillary data layers that are used to modify walking speeds according to ground cover. The model incorporates barrier features as null values which force a detour to find a safe and suitable crossing point. Datasets used are listed in Table 1.

**Table 1.** Data sources.

| Name | Data | Type | Source | Use |
|------|------|------|--------|-----|
| Arctic DEM | 10 m digital terrain model | Raster | https://www.pgc.umn.edu/data/arcticdem/ (accessed on 1 February 2021) | Remoteness, viewsheds, openness, ruggedness |
| LMÍ Landmælingar Íslands | Roads, coastline, buildings, etc. | Vector | https://www.lmi.is/is (accessed on 4 March 2021) | Remoteness, viewsheds, accessibility |
| Landsnet | Power line routes | Vector | https://www.landsnet.is/ (accessed on 19 June 2021) | Viewsheds |
| Open Street Map | Roads | Vector | https://www.openstreetmap.org/ (accessed on 1 February 2021) https://download.geofabrik.de/europe/iceland.html (accessed on 1 February 2021) | Remoteness, viewsheds, accessibility |
| AUI Farmland Database | Land cover | Raster | https://www.moldin.net/nytjaland---aui-farmland-database.html (accessed on 1 February 2021) | Naturalness, remoteness |
| Landscan | Population | Raster | https://landscan.ornl.gov/ (accessed on 22 August 2021) | Accessibility |

### 2.2.2. Absence of Modern Human Artefacts

This attribute refers to the lack of obvious human constructions within the visible landscape, including roads, vehicle tracks, pylons, dams, reservoirs, buildings and other built structures. A subject's feeling of both naturalness and remoteness is significantly affected by the number of human features that are visible at any location within the area of interest and their distance from them. The choice of which human features to include here is driven largely by what is understood to act as a wilderness detractor [30]. Early work on the effects of human artefacts on wilderness quality has tended to focus on simple distance measures [31], with more recent work using measures of visibility of human artefacts derived from viewshed analyses and digital terrain models [22] to calculate the area from which a given artefact can be seen using line-of-sight from one point of a terrain surface to another [32]. A similar approach to that used by Carver et al. [22,25] is adopted here using artefacts that are deemed to have an impact on wilderness, together with a detailed digital surface model (DSM) and a rapid viewshed assessment method developed for the earlier Cairngorm wildness mapping project [33].

It has been shown that the reliability of viewsheds produced in GIS is strongly dependent on the accuracy of the terrain model used and the inclusion of intervening features (buildings, woodland, etc.) or terrain clutter in the analysis [34]. Modern human artefacts are extracted from appropriate datasets (see Table 1) and assigned appropriate height values reflecting how tall they are and, therefore, how prominent they appear in the landscape. Roads are modelled with a 3 m height value used to represent an average vehicle height. Cumulative viewsheds, weighted according to artefact type and distance, are produced using the Viewshed Explorer tool [32] to show the relative effects associated with the presence and absence of human artefacts, and the results processed in ArcGIS Pro 3.0. Bishop's work [35] on the determination of thresholds of visual impact were used to help define the limits of viewsheds and the distance decay function used.

An inverse square distance function is used in calculating the significance of visible cells in the GIS database. This function gives the relative area in the viewer's field of view that a cell or feature occupies in comparison to the background terrain surface taking distance decay effects and the intervening terrain into account. The output is a unitless grid, the numbers in which are dependent on the area of terrain and input features visible from any point on the terrain surface.

### 2.2.3. Perceived Naturalness of Land Cover

Perceived naturalness is described here as the extent to which land management, or lack of it, creates a pattern of vegetation and land cover which appears natural to the casual observer. Perceptions of wilderness are in part related to evidence of land management activities such as fencing, improved pasture and stocking rates, as well as presence of natural or near-natural vegetation patterns. Here, the AUI Farmland [36] data were used to describe perceived naturalness in the Central Highlands. Aspects of land management are identifiable from national land cover datasets and enables their reclassification using additional input from local experts (including mountain guides and park rangers) into the naturalness classes shown in Table 2.

To account for the influence that patterns of land cover within the area immediately around the observer location has upon perceived naturalness, the mean naturalness class is calculated for each location within a 250 m radius neighbourhood using the Focal Statistics tool in ArcGIS Pro 3.0. This unitless value is then assigned to the target cell to represent the overall naturalness score for that location.

### 2.3. WQI and Zone Definition

A simple weighted linear summation MCE model is used to combine all three wilderness attributes into a final WQI. All input attribute layers are normalised onto a common unitless scale that enables cross comparison. This is accomplished by rescaling values onto a 1–256 scale (256 values) using the equal intervals option in ArcGIS Pro 3.0 Slice

tool, where low values are indicative of lower wildness. These normalised values are then applied using an equally weighted MCE analysis within the ArcGIS Pro 3.0 Raster Calculator. This allows the effects of each value to be accounted for and a final value for wildness calculated. Weighting of individual attribute layers may then be altered to account for different perceptions on priorities attached to each attribute but are maintained as equal in this exercise assuming each input layer to the model is of equal importance.

**Table 2.** Naturalness classifications applied to AUI Farmland Data.

| Naturalness Class | Land Cover Class (from AUI Farmland Database) |
|---|---|
| 0 | No Data |
| 1 | Built |
| 2 | Cultivated Land/Shrubland |
| 3 | Grassland/Unknown (Lowland Vegetated) |
| 4 | Rich Heathland/Poor Heathland |
| 5 | Mossland/Damp Wetland/Wetland/Poorly Vegetated/Barren/Lakes/Glacier/Unknown |

This is a continuous model that ranges from least to most wild, and while useful as an indication of these internal patterns, it needs to be reclassified into zones for it to be useful in a planning and policy context for supporting decisions about protected area boundaries. The WQI is therefore reclassified into Interior Core, Core, Buffer and Transition zones based on a Jenks "Natural Breaks" Classification model. This follows the approached used by SNH in their 2014 Phase 2 map of Wild Land Areas in Scotland [23]. The method examines the distribution of the WQI values across the mapped area and divides these into a specified number of classes such that the difference from the mean within each class is minimised. The classification used here uses 5 classes as per the SNH 2014 methodology, with class 5 being labelled 'Interior Core', class 4 as 'Core', class 3 as 'Buffer', class 2 as 'Transition' and class 1 being 'Not Wild'. The Wild Europe Working Definition for wilderness areas is used to identify 'Core' and 'Core plus Contiguous Buffer' areas larger than 3000 ha (30 km$^2$) and >10,000 ha (100 km$^2$), respectively [28]. Jenks class 3 areas not contiguous with 'Core' areas > 3000 ha (together with any class 4 areas < 3000 ha) are classified as 'Buffer' and all class 2 areas as 'Transition' zones. All class 1 areas are classified as 'Not wild'.

### 2.4. Wilderness Character

The wilderness zones derived using the above classification are further classified according to a range of variables describing their geographical nature and wilderness character. This includes area, elevation range, openness, ruggedness, accessibility, mobile phone coverage, livestock grazing and landscape character classes. Further spatial models are needed to map openness, ruggedness and accessibility to centres of population.

#### 2.4.1. Openness

Openness follows the method developed by Yokoyama et al. [37] as a measure to display surface features on a terrain model using a method independent of a light source and as an alternative to other methods such as hillshading. The method allows for the enclosure of each cell to be represented graphically, thus differentiating between wide open spaces and closely enclosed valleys, assisting in defining the openness characteristics of each identified wildland area. Topographic Openness is calculated from the terrain model using the Skyview tool within the QGIS SAGA toolbox. This generates values representing the proportion (percentage) of visible sky for each cell within the dataset.

### 2.4.2. Ruggedness

Ruggedness is taken to refer to the physical characteristics of the landscape including effects of steep and rough terrain that is frequently found across the Central Highlands. A terrain model is used to derive indices of terrain complexity based on total slope curvature (rate of change of slope in both plan and profile). Areas where curvature changes frequently are identified because they are deemed to represent rapidly changing terrain and hence ruggedness. A simple index defined as the standard deviation (SD) of total terrain curvature within a 250 m radius of the target location is used to map variations in terrain ruggedness utilizing the Curvature and Focal Statistics tools in ArcGIS Pro 3.0.

### 2.4.3. Accessibility

While there is a relatively well-developed network of gravel roads across parts of the Central Highlands, with corresponding effects on remoteness from mechanised access as described in Section 3.2, much of Iceland's interior has a remote feel due in part to the time it takes to get there from the main centres of population. This is an essential aspect of the Central Highlands' wilderness character and is modelled here using a population-weighted accessibility surface taking the road network, road type and average speed of driving into account. A combination of a Cost Distance surface calculated using the Distance Accumulation tool in ArcGIS Pro 3.0 and a simple weighted linear summation model in the Raster Calculator is used with centres of population extracted from LandScan global population data. Here we use population density thresholds ($n = 10$) to identify a range of population centres from farmsteads and villages to major towns and the city of Reykjavik. These are used as journey source locations (origins) for the Cost Distance calculations based on average estimated driving speeds according to road type and a background offroad walking speed of 5 km/h. This enables the calculation of isochrone surfaces providing a 'time taken to travel' surface for each of the population density thresholds which are then combined using the Raster Calculator in a linear weighted summation model using the relative population thresholds as weights.

Maps from other existing sources are used to derive wilderness character information pertaining to mobile phone coverage, livestock grazing and landscape character assessments. Mobile phone coverage is remarkably good across much of Iceland, including the Central Highlands. This is an important additional factor influencing wilderness character since it affects the sense of remoteness. The ability to make an emergency call to summon help should it be needed (e.g., in case of personal injury, vehicle breakdown, navigational error, etc.) along with access to digital maps and GPS location has a significant impact on wilderness character, self-reliance, solitude and risk. Livestock grazing is carried out over the summer in parts of the Central Highlands. This includes both sheep and horses, the latter being used principally for recreation. Associated with this grazing activity is fencing, 4x4 tracks and small huts/shelters. As a human economic land use, grazing of animals and associated infrastructure has an influence on wilderness character in the areas where it takes place. Finally, landscape character has been mapped across Iceland and the 27 different landscape type units across 7 categories described in a recent report prepared by EFLA and Land Use Consultants, Scotland [38]. The boundaries of these landscape units and the information contained in the report are used here to supplement the information wilderness character.

## 3. Results

Results from the analysis and models applied are presented as a series of three normalized and unitless wilderness quality attribute maps. These are combined to create a WQI which is in turn classified into wilderness zones and a series of seventeen separate wilderness areas meeting the criteria for European wilderness areas. Three wilderness character maps are also presented to illustrate how further spatial data models can be used and combined with existing maps to describe the unique characteristics of each of the seventeen wilderness areas.

### 3.1. Remoteness from Mechanised Access

Remoteness from mechanised access is calculated here using the above methods described in Section 2.2.1 for both summer and winter conditions to account for differences that occur between the two main seasons. During the summer months, vehicles are restricted to established roads, and off-road driving is specifically prohibited. However, during the winter months these rules are relaxed, and except for some restricted areas, vehicles may travel anywhere in Iceland provided there is sufficient snow and ice cover. The difference in relative remoteness between walking (summer or winter) and off-road driving in 4 × 4 "super jeep" vehicles (winter) is very noticeable, with these vehicles being able to cover greater distances in shorter times. This has potentially far-reaching implications for the designation of areas of IUCN Category 1b wilderness, as described later in the paper. Both summer and winter remoteness surfaces are shown in Figure 3.

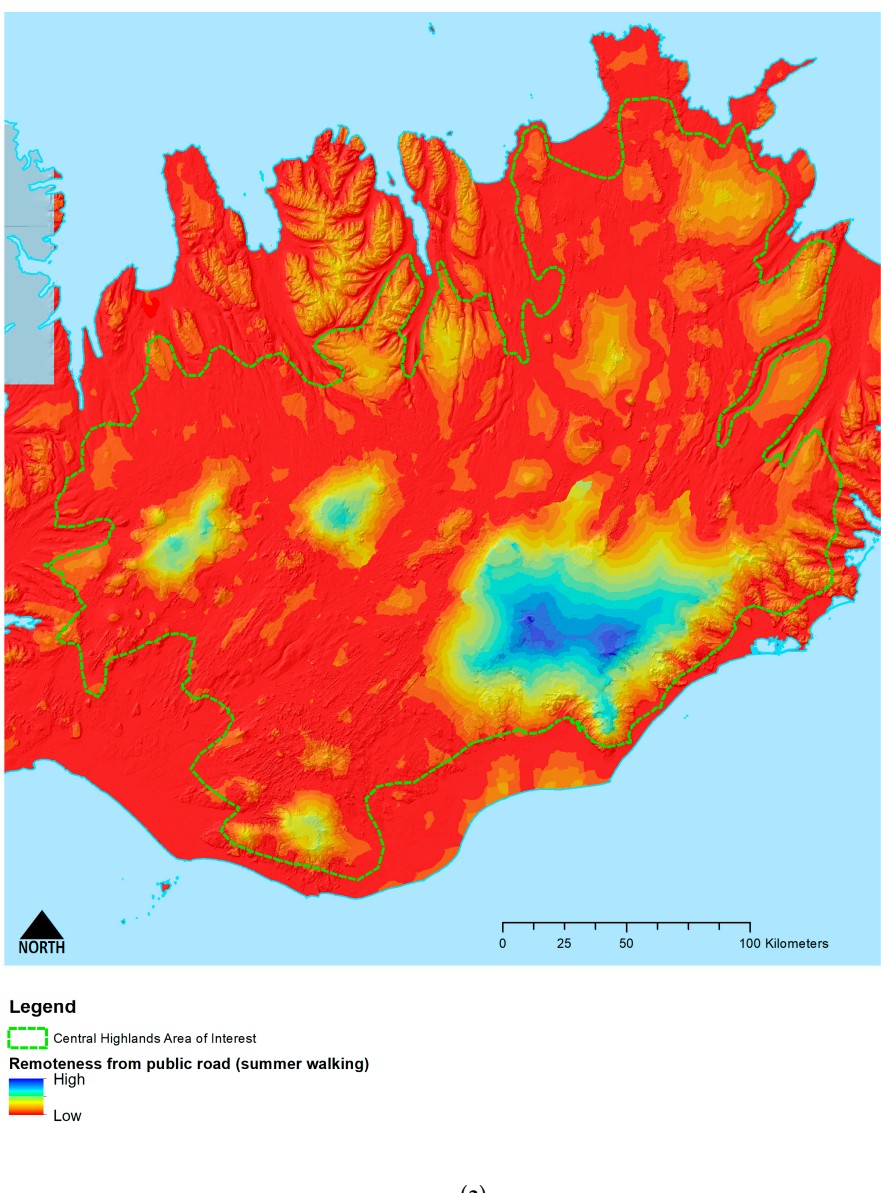

(**a**)

**Figure 3.** *Cont.*

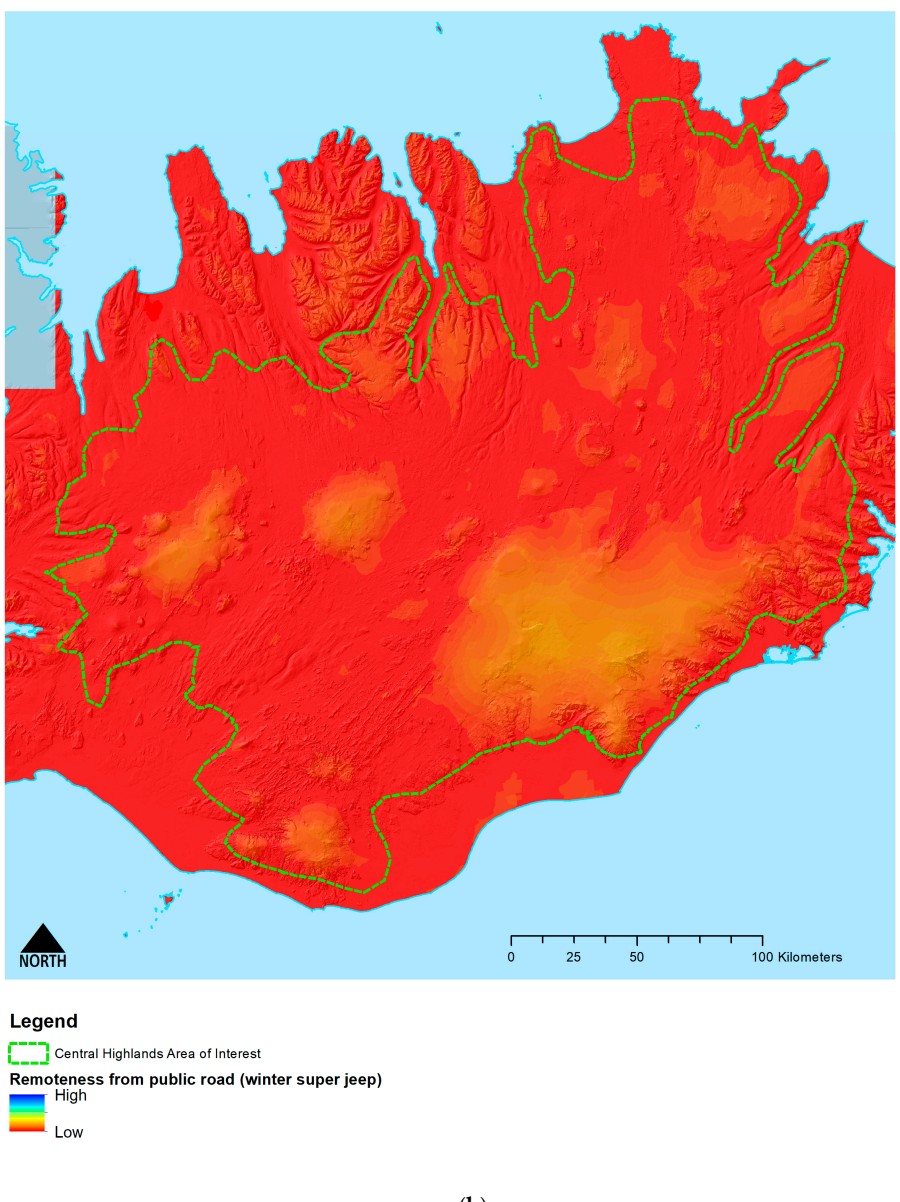

(**b**)

**Figure 3.** Summer (**a**) and winter (**b**) remoteness surface.

### 3.2. Absence of Modern Human Artefacts

Absence of modern human artefacts is used to represent the degree of visual intrusion from built structures in the landscape. The model additionally highlights areas which are in total shadow from all visual features owing to the shape of the local landscape. Such areas of zero visual intrusion from modern human artefacts currently comprise a significant portion of the core areas of the Central Highlands, many of which occupy the interior and valleys which are entirely shielded by their topography. While occurring less frequently in the proximity of modified areas, pockets entirely bereft of visual intrusion can be found everywhere, owing to the high relief and general ruggedness of the terrain. The output layer describing the absence of modern human artefacts, including buildings and other structures, roads, hydro-power schemes and power lines, is shown in Figure 4, with areas of zero visual intrusion highlighted in white.

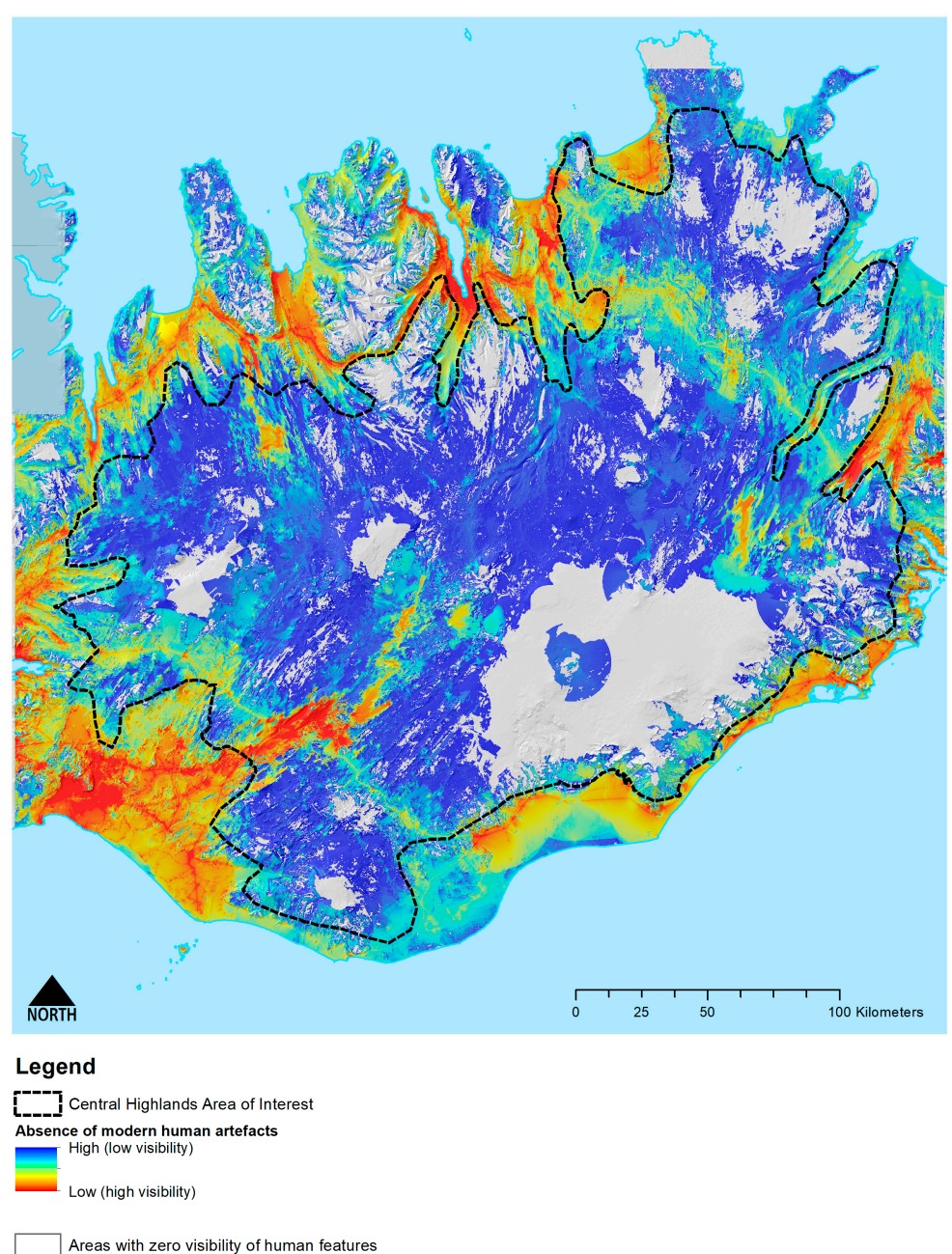

**Legend**

- - - Central Highlands Area of Interest

**Absence of modern human artefacts**

High (low visibility)

Low (high visibility)

Areas with zero visibility of human features

**Figure 4.** Absence of modern human artefacts.

### *3.3. Perceived Naturalness of Land Cover*

Perceived naturalness of land cover is mapped from the AUI Farmland Database using the methods described in Section 2.2.3. The resulting attribute map is shown in Figure 5. Except for the areas immediately surrounding roads, huts, reservoirs and associated power infrastructure, the vast majority of the Central Highlands presents as the highest category on the naturalness scale. The effects of farming and urban areas around the coast fringe are clearly visible in the lower naturalness scores seen in these regions.

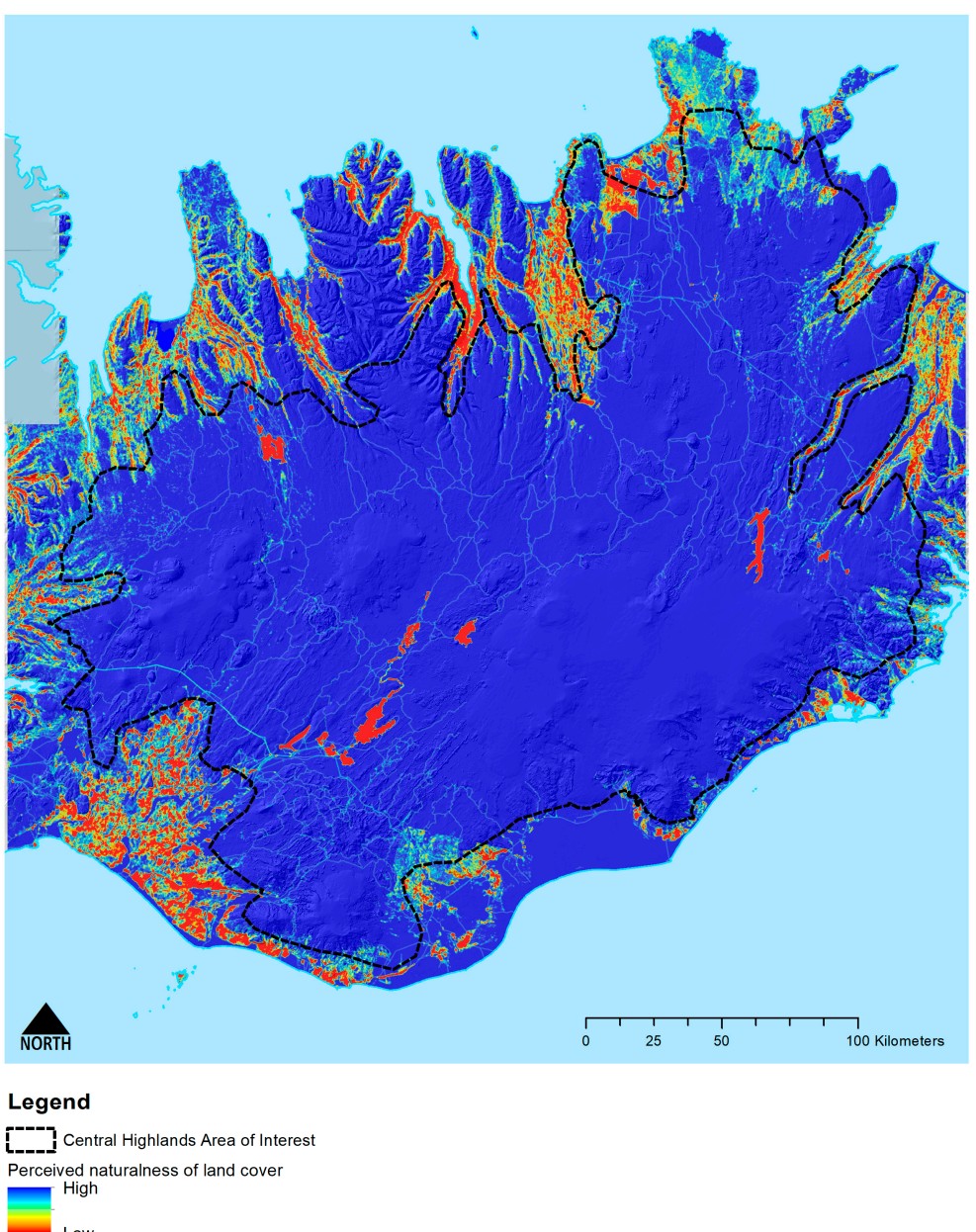

**Figure 5.** Perceived naturalness of land cover.

*3.4. Wilderness Quality Index*

The final WQI is shown in Figure 6. This shows the pattern in spatial variations in wilderness quality across the whole of the Central Highlands study area taking the three wilderness attributes of remoteness, visual impact from human features and naturalness of land cover into account. A series of five wilderness zones based on the reclassification of the WQI is shown in Figure 7. Strong spatial patterns influenced by the major icecaps of the Vatnajökull, Hofsjökull, Langjökull and Mýrdalsjökull can be seen as defining the Interior Core wilderness zones and the network of gravel roads, powerlines, hydro-power schemes and other human infrastructure playing a major role in defining the pattern of buffer and transition zones. Hydro-power reservoirs are large unnatural features and so stand out particularly strongly in Figure 6. Roads and power lines emanating from these complete the picture, dissecting the Central Highlands area into a series of large wilderness areas (Core and Interior Core zones) and their surrounding Buffer and Transition zones in Figure 7.

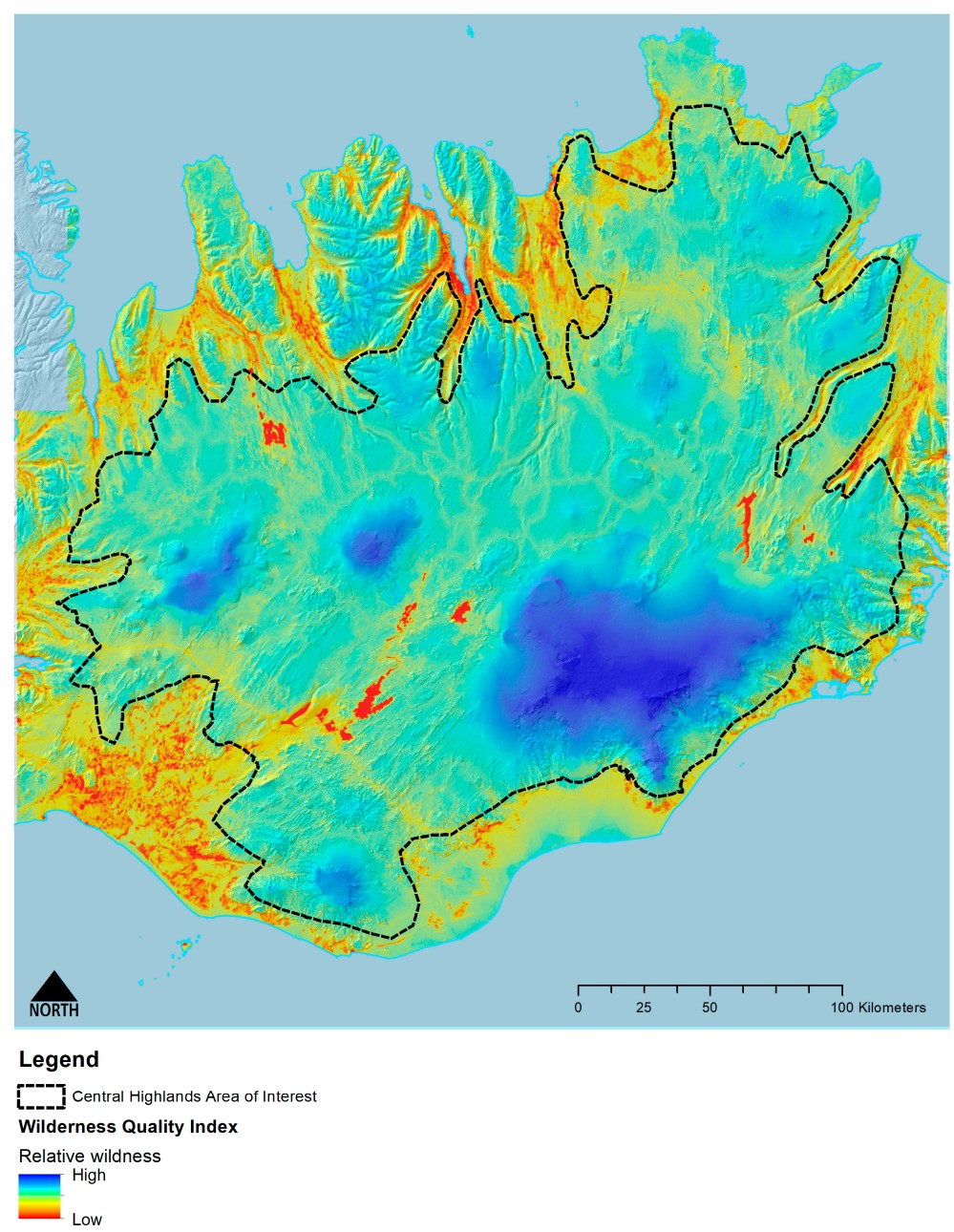

**Figure 6.** Wilderness Quality Index (WQI) for the Central Highlands.

*3.5. Wilderness Area Definition*

Applying the size/area constraints from the Wild Europe Working Definition identifies Core wilderness zones as Interior Core and Core areas (Jenks classes 4 and 5) larger than 3000 ha (30 km$^2$) together with contiguous buffer zones (Jenks class 3) larger than 10,000 ha (100 km$^2$) as wilderness. These are shown in Figure 8 together with core areas less than the required 3000 ha and transition zone (Jenks class 2) as possible IUCN Category 2 areas. This results in the delineation of seventeen wilderness areas across the Central Highlands and adjacent landscapes. Of these, fourteen lie inside the Central Highlands and three outside, totaling some 28,470 km$^2$, of which 26,404 km$^2$ is inside and 2066 km$^2$ is outside the area of interest. Together, these cover over 47 percent of the Central Highlands area of interest (55,400 km$^2$), plus three in adjacent areas, of which 19,500 km$^2$ is public land and 8970 km$^2$ privately owned. Also shown on this map are the existing protected areas. These include the internationally important Vatnajökull National Park, the Mývatn-Laxá and Þjósárver Ramsar Sites and the Þjórsáver and Fjallabak Nature Reserves, but crucially in respect to

the work and results presented here, there are no extant designated wilderness areas. While these wilderness areas are geographically distinct, some are divided and fragmented by narrow corridors created by gravel roads, further illustrating the significance of mechanised access on remoteness and visual impact.

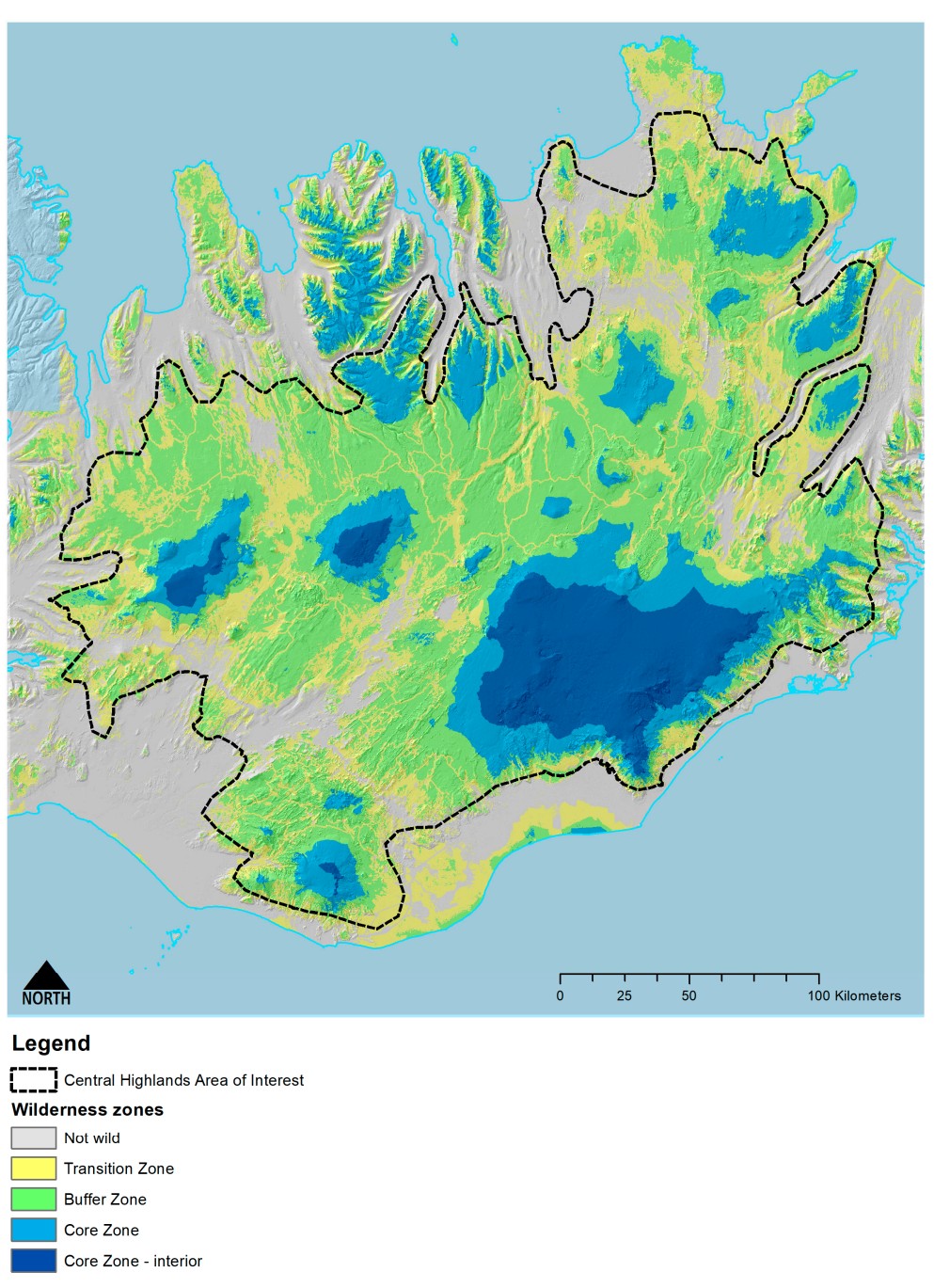

**Figure 7.** Wilderness zones in the Central Highlands.

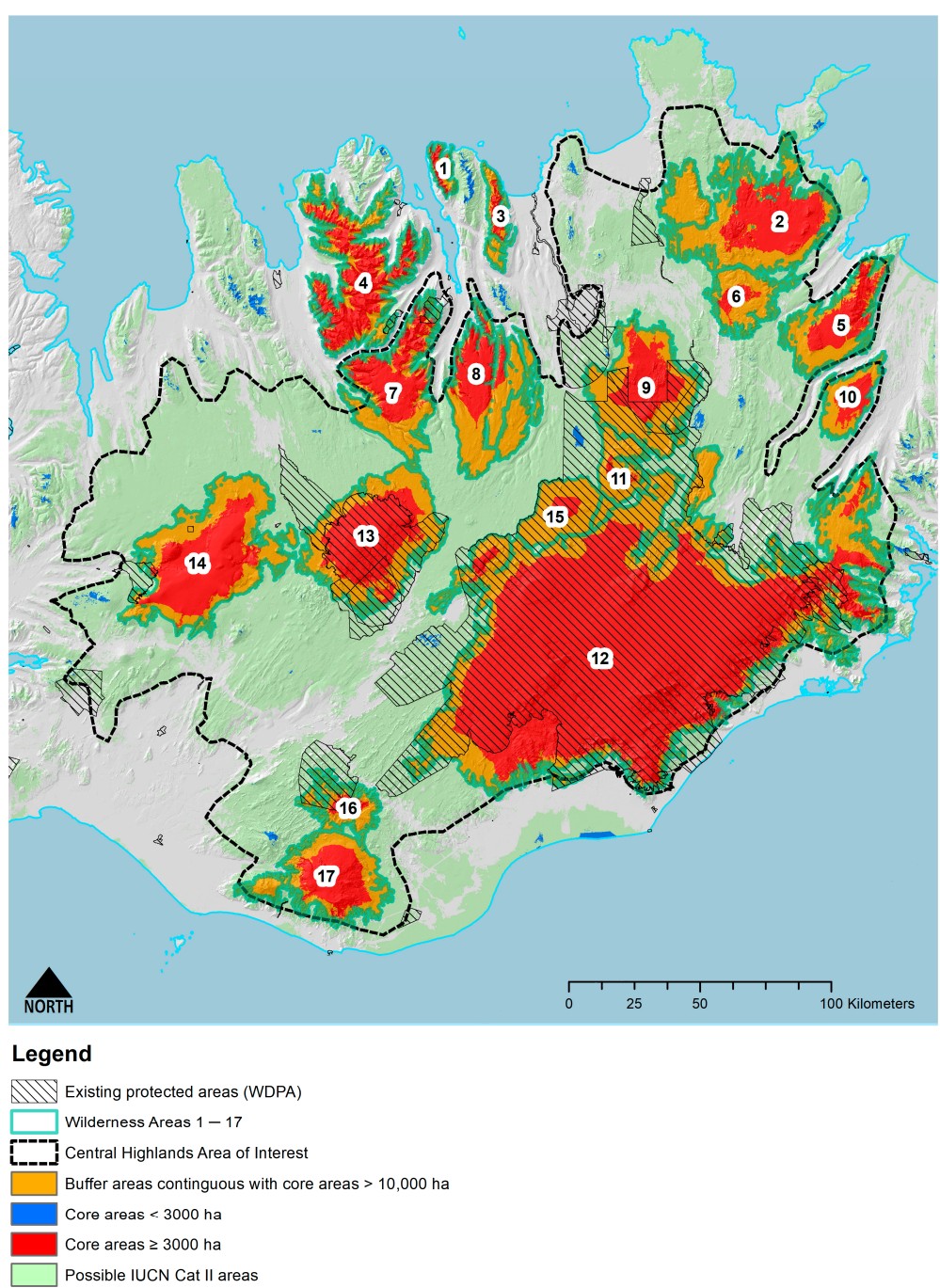

**Legend**

- ⧄ Existing protected areas (WDPA)
- ▭ Wilderness Areas 1 — 17
- ⬚ Central Highlands Area of Interest
- 🟧 Buffer areas continuous with core areas > 10,000 ha
- 🟦 Core areas < 3000 ha
- 🟥 Core areas ≥ 3000 ha
- 🟩 Possible IUCN Cat II areas

**Figure 8.** Wilderness areas 1–17 meeting Wild Europe Working Definition.

### 3.6. Wilderness Character

The wilderness areas shown in Figure 8 are further classified according to a range of variables describing their geographical nature and wilderness character, including the modelled and normalized variables for openness, ruggedness and accessibility as shown in Figures 9–11. Table 3 summarises each of the seventeen wilderness areas by their geographical characteristics. The character of each wilderness area is described in further detail. Area 12 Vatnajökulssvæðið is provided here as an example (see Figure 12).

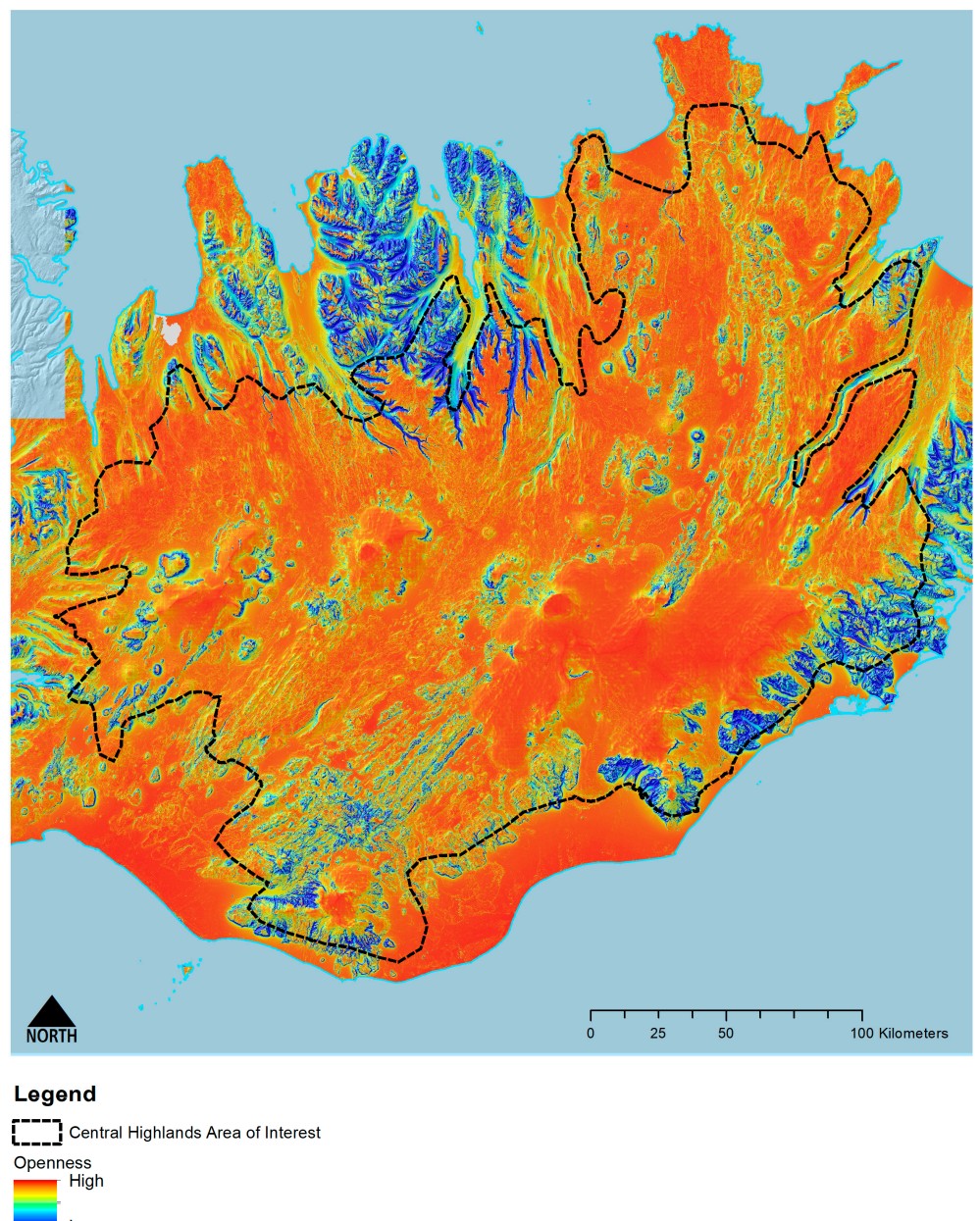

**Figure 9.** Obtained values for Openness for the Central Highlands.

**Table 3.** Wilderness character summary figures.

| No [1] | Name | Area (km$^2$) | Altitude (m) | Openness (Mean, %) | Ruggedness [2] (Mean) | Accessibility [3] (Mean) |
|--------|------|---------------|--------------|--------------------|-----------------------|--------------------------|
| 1 | Keflavík og Látraströnd | 124 | 17–1168 | 88 | 1.54 | 22,180 |
| 2 | Heljardalsfjöll | 2083 | 30–983 | 97 | 0.40 | 30,213 |
| 3 | Náttfaravíkur og Kinnarfjöll | 237 | 9–1214 | 91 | 1.11 | 20,507 |
| 4 | Tröllaskagi | 1478 | 34–1440 | 89 | 1.33 | 18,167 |
| 5 | Smjörfjöll | 870 | 109–1255 | 96 | 0.53 | 29,108 |
| 6 | Dimmifjallgarður | 511 | 351–1037 | 96 | 0.52 | 25,968 |
| 7 | Nýjabæjarfjall | 1198 | 189–1541 | 93 | 0.93 | 19,060 |
| 8 | Bleiksmýrardalur | 1402 | 130–1254 | 96 | 0.62 | 20,225 |
| 9 | Ódáðahraun | 1379 | 382–1678 | 98 | 0.44 | 29,226 |
| 10 | Fljótsdalsheiði | 413 | 297–710 | 99 | 0.25 | 29,548 |
| 11 | Askja í Dyngjufjöllum | 380 | 523–1517 | 96 | 0.60 | 29,530 |
| 12 | Ríki Vatnajökuls | 12,315 | 4–2108 | 97 | 0.53 | 30,002 |

**Table 3.** *Cont.*

| No [1] | Name | Area (km²) | Altitude (m) | Openness (Mean, %) | Ruggedness [2] (Mean) | Accessibility [3] (Mean) |
|--------|------|-----------|--------------|--------------------|-----------------------|--------------------------|
| 13 | Hofsjökull og Þjórsárver | 1907 | 554–1789 | 98 | 0.35 | 18,796 |
| 14 | Langjökull | 2095 | 294–1670 | 97 | 0.45 | 14,472 |
| 15 | Trölladyngja | 546 | 750–1465 | 98 | 0.38 | 25,674 |
| 16 | Fjallabak | 408 | 67–1383 | 93 | 1.26 | 14,115 |
| 17 | Mýrdalsjökull og Eyjafjallajökull | 1124 | 56–1637 | 95 | 0.87 | 13,426 |

[1] Number code for each of the seventeen wilderness area corresponding to the numbers and locations shown in Figure 8. [2] Ruggedness is a unitless number calculated as standard deviation of slope curvatures (rate of change of slope) within a 250 m radius. Higher numbers indicate greater ruggedness. [3] Accessibility is a unitless number calculated as a population and distance weighted surface taking typical road class driving speeds into account. Lower numbers indicate an area closer to more populated areas, such as Reyjavik and Akureyri (with shorter driving times), and higher numbers indicate those further away (with longer driving times).

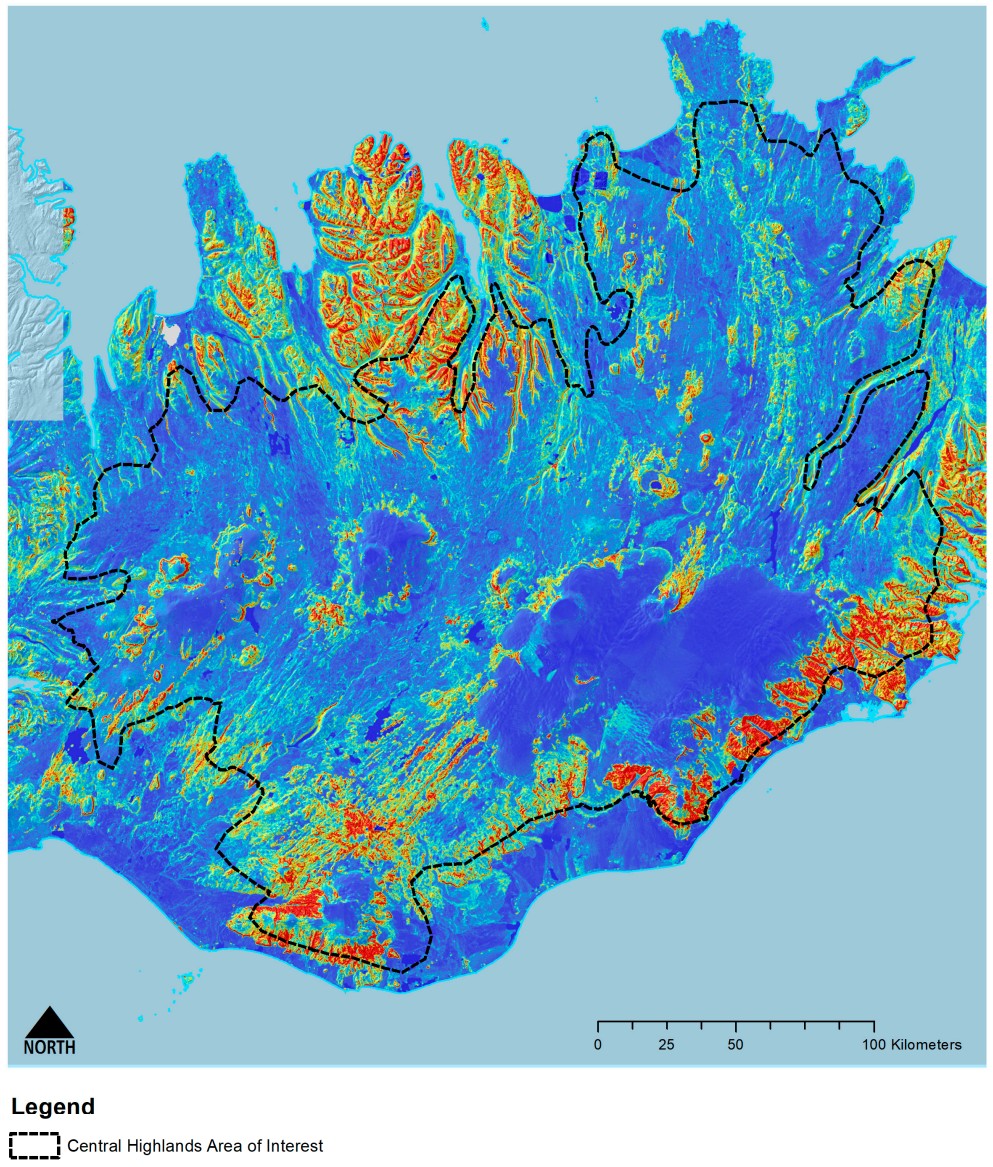

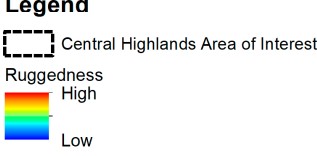

**Legend**

- - - - Central Highlands Area of Interest

Ruggedness

High

Low

**Figure 10.** Obtained values for Ruggedness for the Central Highlands.

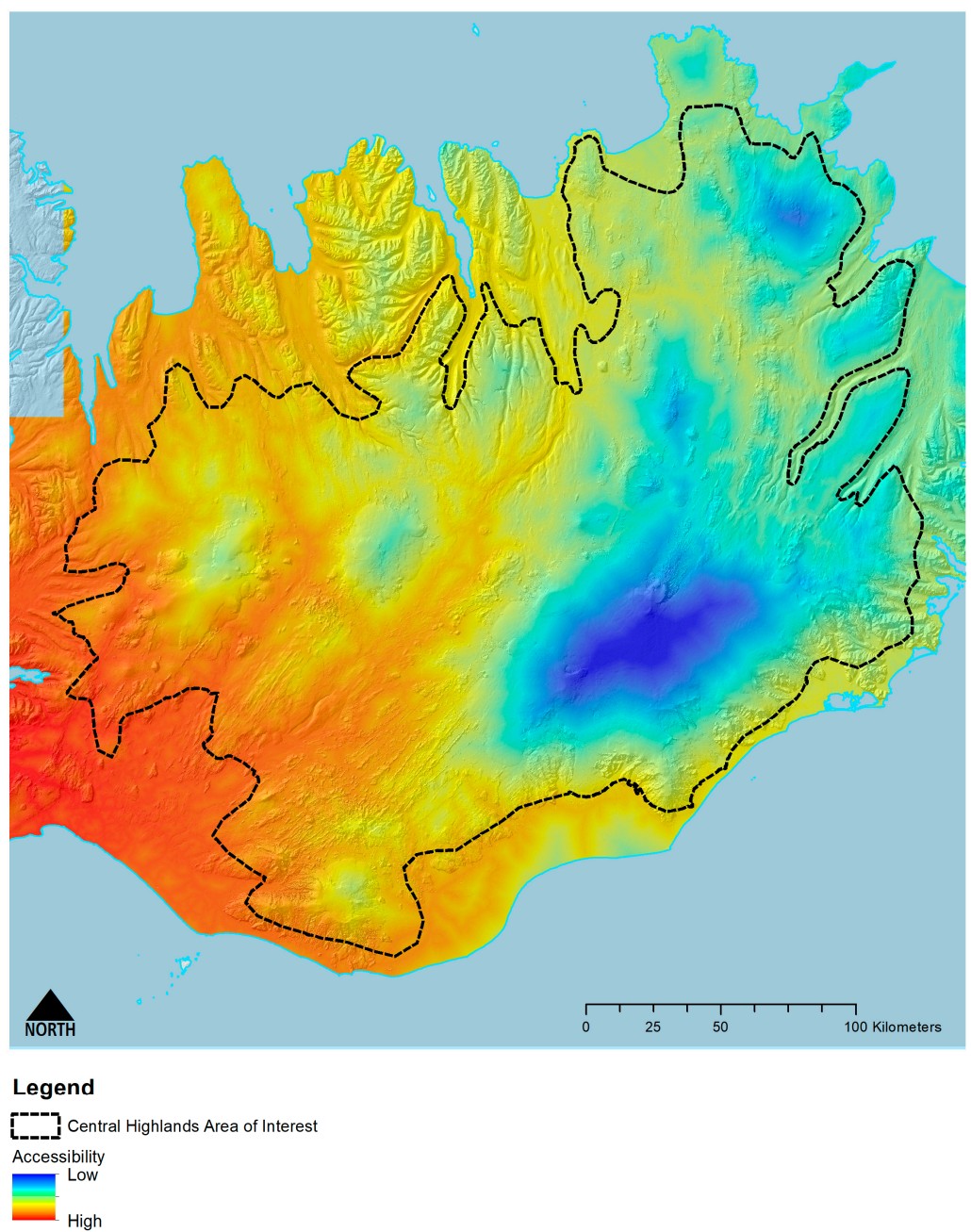

**Figure 11.** Obtained values for accessibility for the Central Highlands.

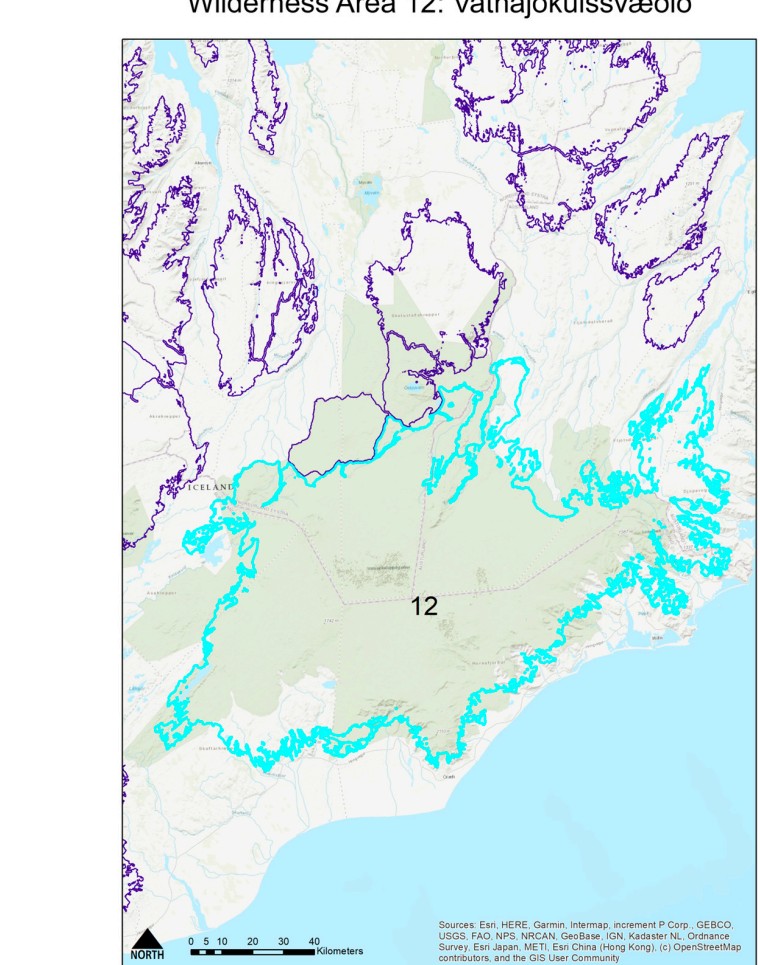

## Wilderness Area 12: Vatnajökulssvæðið

**General setting and description:** The Vatnajökull and its surrounding landscape is perhaps the most iconic of all Icelandic wilderness landscapes. The area is large and is the largest ice cap in Iceland.

**Topography** The topography of Vatnajökulssvæðið is dominated by the vast ice cap, sub-glacial volcanoes (Grímsvötn, Bárðarbunga, Tungnafellsjökull, Kverkfjöll and Öræfajökull), glacier flows from around its edges and rugged mountain ridges. The Vatnajökulssvæðið also includes an area of coastal mountains and uninhabited valleys in the east.

**Landscape assessment:** The landscape is varied with large expansive open areas on the ice cap and to the north and complex, enclosed, and rugged areas around the edges of the glacier to the south and east. Here, the ice flows have carved deep valleys with further open areas across the expansive lakes and rivers along its southern coastal margin.

**Land use:** Most of the landscape is snow- and ice-covered and there is no livestock grazing to the west and north of the glacier, but sheep graze in the southeast valleys where there is also reindeer hunting. The main land use is recreation and tourism. Mobile phone coverage is generally poor, with many areas without signal. Much of the area is extremely remote.

**Figure 12.** Vatnajökulssvæðið wilderness character and description.

## 4. Discussion

The use of proxy measures for wilderness area mapping has its origins in some of the earliest global scale mapping. McCloskey and Spalding [31] defined the world's remaining wilderness as those areas more than six kilometres from the nearest settlement, road, railway or navigable river using 1:2 million scale Jet Navigation Charts. Ibisch et al. [39] provide a more up-to-date estimate of the world's remaining roadless areas using a buffer distance of 1 km, finding that only 7% of the world's land surface is covered by roadless areas greater than 100 km$^2$. While such buffers are useful as global proxies, remoteness and visual impact are better modelled using more sophisticated methods at national or local scales. For example, is it safe to assume that all roads are equal? Does a paved highway exert a greater influence than a gravel track? Does a small cluster of farm buildings have the same impact as a large town or city? How does topography and associated barriers to movement and resistance to travel affect their impact? Does the fact that you can or cannot see the nearest road from where you stand alter how you think about remoteness? All these factors and their influence are too complex to map using simple buffer zones and thus require more nuanced models that measure their impact in terms of remoteness and visibility.

It is instructive to compare the wilderness areas in Figure 8 with previous wilderness maps drawn for Iceland. These include the EU Wilderness Index [9], the map provided by Ólafsdóttir and Runnström [40], and the most recent map by Ostman et al. [20]. Figure 13 shows these maps superimposed over the seventeen wilderness areas from Figure 8. A simple visual comparison of the wilderness areas developed here and those based on the EU-level WQI from Kuiters et al. [9] in Figure 13a demonstrates a reasonable degree of similarity. This is only to be expected since, despite differences in criteria, data and approach, these maps are dealing with the same landscape and the same underlying characteristics of wilderness, namely, remoteness and naturalness, measured along a continuum from least to most wild. Comparisons with those maps derived from simple buffer zones around selected human features show much larger levels of disagreement, with the maps from Ólafsdóttir and Runnström [40] and Ostman et al. [20] including substantially greater areas of wilderness when compared to the results of the current analysis.

The Ólafsdóttir and Runnström [40] map in Figure 13b is a straightforward spatial mapping of the criteria described in the previous text of the NCA No 44/1999, which maps those areas more than 5 km from a road or building as simple buffers and then selects those that are more than 25 km$^2$ in size. Here, all buildings and public roads are used regardless of road grade or building size, with the result that a small hut or shelter has the same effect as a large geothermal power station on the wilderness buffers. The scale of development and the influence or impact that this has on the landscape is therefore not considered. The work by Ólafsdóttir and Runnström [40] does expand the mapping further by including a binary viewshed analysis to show the zones of theoretical visibility (ZTVs) of human features, but this is not included in the final wilderness map.

The Ostman et al. [20] map shown in Figure 13c employs the same criteria but excludes gravel roads from consideration, despite their proven impact on remoteness and visibility. Previous work by Árnason et al. [41] applied the 5 km buffer to all roads in the national register of the Road Authority, producing a map that is much nearer to that by Ólafsdóttir and Runnström [40]. Ostman et al. [20] apply buffers of 3 km and 5 km around power lines depending on the voltage level. There is an attempt to take relative level of impact into account by varying the buffer distances applied based on a scoring system calculated from the use and number of buildings/structures present, their surface area, visibility and connection to the road network, while paved roads are buffered at a uniform 5 km. The resulting wilderness area boundaries are much more extensive than those presented by Ólafsdóttir and Runnström [40] or in the work presented here and conform more closely to the suggested IUCN Category 2 areas shown in Figure 8. This is largely due to the exclusion of gravel roads from consideration and the use of simple buffering, albeit modified with a scoring system.

The exact boundaries of the core areas and buffer/transition zones drawn here in Figure 8 are, in contrast, derived from detailed spatial data and models that measure the impact of human artefacts, remoteness and naturalness to create a WQI rather than relying on simple proxies such as distance buffers. The WQI is classified using statistical methods that take the full range of wilderness quality measures across the Central Highlands into account. As a result, the boundaries at this stage tend to be complex and quite fragmented as seen in Figure 8. It is suggested here that these will need to be simplified for planning and policy use (as with the Phase 3 WLA boundaries produced by SNH [23]), but that the maps provide a rigorous and robust approach to informing such policy decisions at a later stage in the designation process.

The reliability and repeatability of the methods developed here naturally lend themselves to "what if?" analyses of proposed future developments. This, again, can provide an invaluable source of information to support planning and policy decisions regarding development proposals for significant infrastructure within or adjacent to wilderness areas. Such repeat modelling of wilderness quality with and without the features in place can be used to gauge the impact of the proposed development and quantify the area of wilderness lost should the development be allowed to move forward.

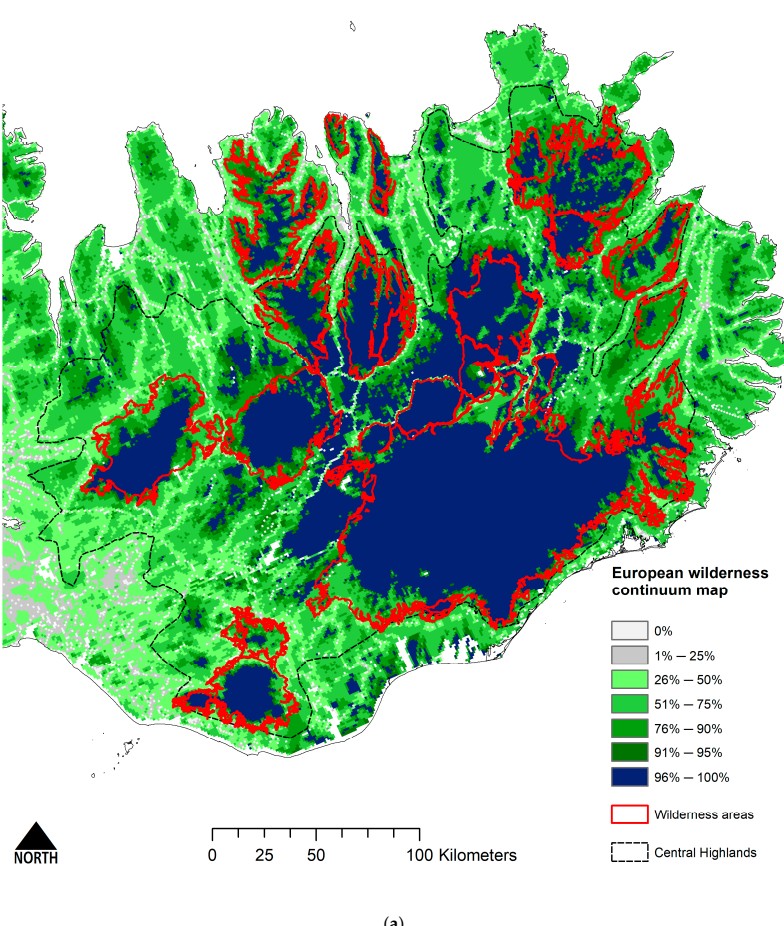

(a)

**Figure 13.** *Cont.*

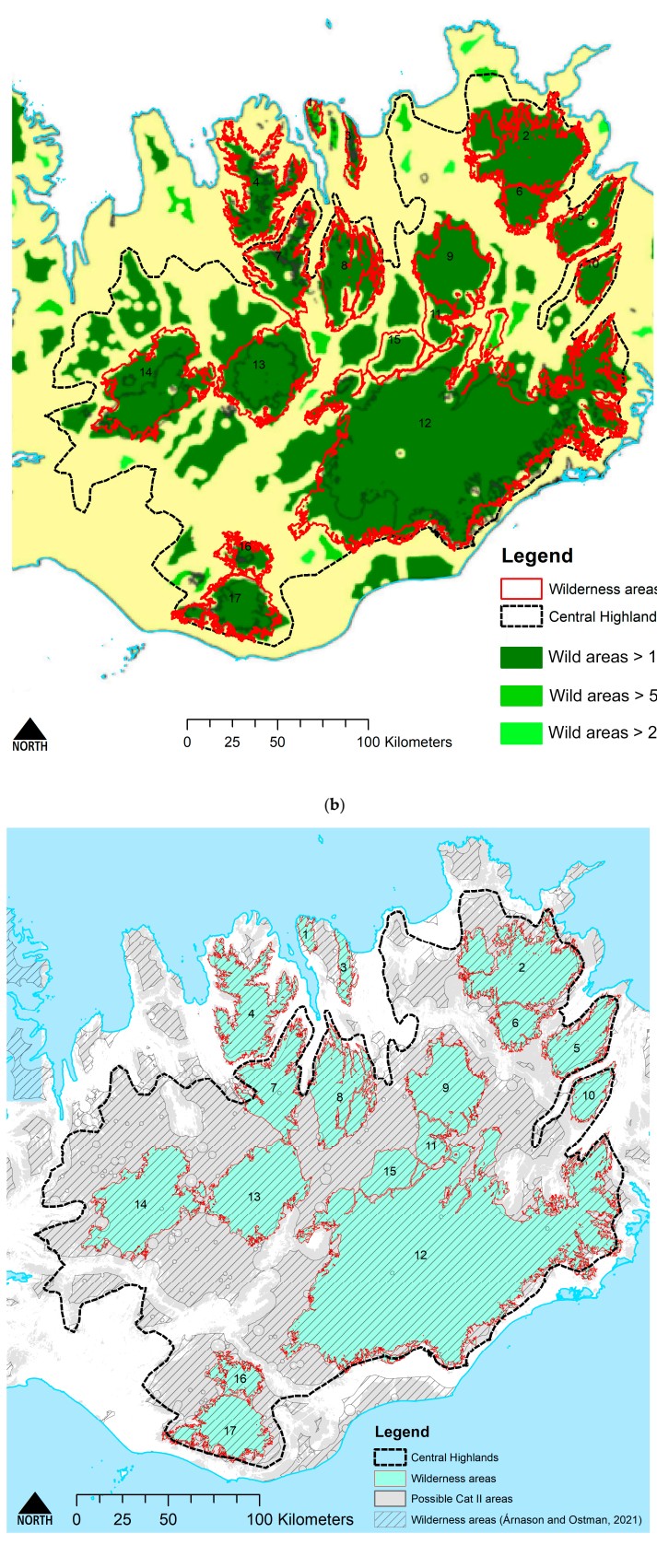

**Figure 13.** Comparison with wilderness maps from: (**a**) Kuiters et al. (2013), (**b**) Ólafsdóttir and Runnström (2011), and (**c**) Ostman et al. (2021).

Winter driving offroad over snow and ice remains an issue that requires further attention. While much of the mapping and analysis carried out here relates to summer conditions and rules (e.g., limiting vehicles to those roads usable by the public), the maps in Figure 3 demonstrate the potential effect of winter offroad driving in greatly reducing remoteness. This is an issue that could potentially limit opportunities for the Icelandic government to designate large areas of the Central Highlands under IUCN Category 1b due to the explicit exclusion of mechanical means of transportation in IUCN wilderness area guidelines. This requires careful engagement with the 4x4 community to explore options for limiting offroad winter driving to certain areas outside of mapped wilderness cores as mentioned in Article 46(2) of the NCA " ... and to ensure that present and future generations can enjoy solitude and the nature without disturbance from man-made structures or the traffic of motorized vehicles".

## 5. Conclusions

The co-related aims of protecting pristine nature and facilitating tourism and recreational use is a key challenge facing the Icelandic government in the Central Highlands. This requires striking a careful balance between visitor use, resource exploitation and the preservation of nature [42]. Nowhere is this more important than in the potential conflicts between winter offroad driving, renewable energy developments and wilderness designation. Detailed and accurate mapping of landscape attributes and human impacts are key to sustainable decision making about wilderness landscapes in this regard. This paper presents a significant improvement on existing approaches to mapping wilderness areas in Iceland both in terms of detail and methods used and one that carefully considers and takes account of local nature conservation legislation.

The work described is the most detailed and accurate mapping of wilderness quality and wilderness character for the Central Highlands of Iceland that has been carried out to date. This has enabled the definition of seventeen separate and distinct wilderness areas along with surrounding buffer and transition zones. A key advantage over existing studies is the use and adaptation of internationally recognised methods and wilderness standards which use direct measurement and modelling of spatial factors determining wilderness quality. This is supplemented by wilderness character assessments based on additional mapping and descriptions of spatial factors affecting the individual wilderness landscapes and their unique character. The use of a 4Rs approach ensures rigour, robustness, repeatability, and reliability in the work carried out.

The work and the maps presented in this paper differ significantly from previous work in that rather than using simple distance/area proxies, the attributes mapped here represent the actual measurement of human impacts from land use, settlement, and infrastructure development on wilderness landscapes. The WQI and seventeen wilderness areas identified can be seen as an important step towards the formal definition of boundaries of wilderness areas meeting IUCN Category 1b and Wild Europe Working Definition in Iceland. Further work is recommended to complete the mapping for the whole of Iceland as mandated in the amendment to the 2013 NCA in Article 73a 2021 [11]. This could be supplemented where necessary by additional models to account for variations in remoteness around the coastal areas and islands, where different modes of travel/access will play an important role, and by comparison with ecological data on protected habitats and species distributions.

Finally, we suggest that the 4Rs approach developed here, along the methods and models applied, could be usefully applied across all countries of Europe taking the individual national datasets and conditions pertaining to wilderness and its relevance to social, political and cultural understanding into account. This could, with cross-border collaboration where necessary, help better map the patterns of Europe's remaining wilderness areas and inform decisions regarding their future protection in meeting the recommendations from the European Parliament resolution on wilderness [7] and joint agreements on nature protection and restoration of degraded ecosystems under the UN Sustainable Development Goals [5], the Global Biodiversity Framework Convention on Biological Diversity action

oriented targets and the recent Kunming–Montreal agreement calling on signatories to protect 30% of land and sea for nature by 2030 [43]. If we are to meet these commitments, then rigorous, robust, reliable and repeatable methods of mapping wilderness boundaries will be required in supporting the decisions made.

**Author Contributions:** Conceptualization, S.C., S.K. and S.G.; methodology, S.C.; software, S.C.; validation, S.C., S.K. and S.G.; formal analysis, S.C., B.C. and O.K.; investigation, S.C., S.K., S.G., B.C. and O.K.; resources, S.K. and S.G.; data curation, S.C.; writing—original draft preparation, S.C.; writing—review and editing, S.K., S.G. and B.C.; visualization, S.C., B.C. and O.K.; supervision, S.C. and S.K.; project administration, S.C., S.K. and S.G.; funding acquisition, S.K. and S.G. All authors have read and agreed to the published version of the manuscript.

**Funding:** This research was funded by SKH Foundation inc., the Nell Newman Foundation and the Viljandi Foundation.

**Data Availability Statement:** All the model outputs referred to in this paper will be made available through the Icelandic government in due course.

**Acknowledgments:** We gratefully acknowledge the following organisations and individuals for their assistance in developing the work described in the paper: David Ostman, Þorvarður Árnason, Rannveig Ólafsdóttir and Anna Dóra Sæþórsdóttir for providing data on previous mapping work; Veðurstofa Íslands, Landmælingar Íslands, NGA-NSF, Landbúnaðarháskóli, Landsnet, Landsvirkjun, RARIK and OpenStreetMap for access to datasets and expertise; Fanney Ásgeirsdóttir and colleagues in Vatnajökull National Park for local knowledge and field assistance; Snorri Baldursson of Skrauti; and university students Ester Alda Hrafnhildar Bragadóttir, Helga Østerby Þórðardóttir and Finnur Ricart Andrason.

**Conflicts of Interest:** The authors declare no conflict of interest.

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
