# Peer review of "New Approaches to Modelling Wilderness Quality in Iceland"

_land, doi:10.3390/land12020446_

Round 1
Author Response
We would like to thank the two reviewers for their encouraging and helping comments and suggestions. We are replying to these in turn here.
We agree that it would be good to include information on protected habitats. We have therefore added the boundaries of existing designated nature areas from the WDPA in Figure 8 and included a note about these in the discussion. However, we do note that wilderness quality models focus on landscape rather than ecosystem attributes and as such do not include information on biodiversity per se though there are strong relationships between wild(er)ness and biodiversity/naturalness. These are discussed elsewhere but are beyond the scope of the current paper.
We agree that the “what if?” analyses are somewhat outside of the geographical scope of the current paper so have removed this as suggested.
We have simplified the title as requested.
All the figures for wilderness quality attributes and wilderness character maps are shown as normalised values and as such are on a common scale from high to low. We have added extra text in the body of the paper to clarify the nature of these mapped scales.
We have moved all the maps to the results section to keep methods and results separate.
We have added extra text in the methods section to clarify which GIS packages and tools have been used in creating these maps.
We have added clarifications to the figure titles as requested.
Reviewer 2 Report
The study presents a new and interesting method of wilderness delimitation. This, however, may be too little to attract readers who are not very familiar with the problem. The introduction need to be rebuilt in order to present your work in broader perspective. In its actual form, this section is very chaotic and do not respond to the basic questions.
Why this? At least 3 deifnitions, to some extent contradictory, are presented, but you do not refer to them to explain the specificity of your case.
Why now? Partial information are scattered over the introduction, should be gathered and structured.
Why this way? I like the method, but, again, more explanations are needed why you selected these specific descriptors and what are the strongest points of the method.
Why should the reader care? This should be expressed more clearly in introduction and also underline in conclusions. What are the advantages of implementing the proposed method?
In the methods section, add explanation on validation of the results.
The last part of the results section (ca from line 576) should be separated as the discussion. To make it easier to follow, think over the structure of this part.
Figures: There is no need to reprint figures 1 and 2, the reference in the text will be sufficient.
Author Response
We would like to thank the two reviewers for their encouraging and helping comments and suggestions. We are replying to these in turn here.
We have broadened the introduction and the conclusions to present the work more clearly in the broader perspective of global, European and national wilderness mapping programmes.
The introduction has been restructured to make the hierarchy of wilderness definitions from global (IUCN Cat 1b) to European (EU and Wild Europe Working Definition) and national (Icelandic Nature Protection Act) much clearer in how this then applies to the current mapping work and why the work is of interest.
We have restructured the introduction to clarify both the structure of our argument and make clear the logic of why this work is important and topical in respect to GBF CBD targets, EU resolutions and Icelandic NCA policy development.
We have separated the method and results and discussion sections and included clarifications as to why these descriptors both in terms of wilderness quality attributes and wilderness character descriptors are important and appropriate for the work in hand.
We have restructured the discussion to improve the clarity and highlight the advantages of the proposed method over existing approaches, especially in respect to the developing NCA and recent amendments.
We have removed figures 1 and 2 from the Kuiters et al., 2013 report.
Round 2
Reviewer 2 Report
Thank you for restructuring the paper,it is much clearer now.